# Orthogeriatric Management: Improvements in Outcomes during Hospital Admission Due to Hip Fracture

**DOI:** 10.3390/ijerph18063049

**Published:** 2021-03-16

**Authors:** Francisco José Tarazona-Santabalbina, Cristina Ojeda-Thies, Jesús Figueroa Rodríguez, Concepción Cassinello-Ogea, José Ramón Caeiro

**Affiliations:** 1Department of Geriatric Medicine, Hospital Universitario de la Ribera, Alzira, 46600 Valencia, Spain; 2CIBERFES, Centro de Investigación Biomédica en Red Fragilidad y Envejecimiento Saludable, Instituto Carlos III, 28029 Madrid, Spain; 3Department of Orthopaedic Surgery and Traumatology, Hospital Universitario 12 de Octubre, 28041 Madrid, Spain; cristina.ojeda@salud.madrid.org; 4Department of Physical Medicine and Rehabilitation, Complejo Hospitalario Universitario de Santiago de Compostela, 15706 Santiago de Compostela, Spain; jesfi@msn.com; 5Department of Anaesthesiology, Hospital Universitario Miguel Servet, 50009 Zaragoza, Spain; ccassinello@salud.aragon.es; 6Department of Orthopaedics and Traumatology, Complejo Hospitalario Universitario de Santiago de Compostela, 15706 Santiago de Compostela, Spain; jrcaeiro@telefonica.net

**Keywords:** hip fractures, geriatric assessment, orthogeriatric care, functional recovery, geriatric syndromes, mortality, hip fracture surgery, multidisciplinary care

## Abstract

Hip fractures are an important socio-economic problem in western countries. Over the past 60 years orthogeriatric care has improved the management of older patients admitted to hospital after suffering hip fractures. Quality of care in orthogeriatric co-management units has increased, reducing adverse events during acute admission, length of stay, both in-hospital and mid-term mortality, as well as healthcare and social costs. Nevertheless, a large number of areas of controversy regarding the clinical management of older adults admitted due to hip fracture remain to be clarified. This narrative review, centered in the last 5 years, combined the search terms “hip fracture”, “geriatric assessment”, “second hip fracture”, “surgery”, “perioperative management” and “orthogeriatric care”, in order to summarise the state of the art of some questions such as the optimum analgesic protocol, the best approach for treating anemia, the surgical options recommendable for each type of fracture and the efficiency of orthogeriatric co-management and functional recovery.

## 1. Introduction

Osteoporotic hip fractures are one of the main health problems of geriatric patients. Approximately 1.3 million hip fractures were diagnosed in 1990 worldwide [1], and this worldwide annual incidence is expected to increase to over 6 million globally by 2050 [2]. Nearly 80% of the fractures suffered by women and 50% of those in men occur after reaching the age of 70 years [3]. Ninety percent of the fractures occur after falls from standing height [4]. Mortality rates of 10% during acute hospital admission and 30% at one year [5,6] have been reported, but these figures can be reduced introducing an orthogeriatric team [7]. Orthogeriatric care can be defined as the collaboration between orthopedic surgeons and geriatricians to improve hip fracture patient outcomes during hospital admission [8].

Survival after hip fracture does not imply full recovery. Of those who survive, only half recover the functional level they had before the injury [9,10] and one quarter of previously independent patients require admission to an elderly care home [11]. The estimated socio-economic costs derived from treating hip fractures represent 0.1% of global health care costs worldwide increasing to 1.4% in more developed regions [1].

The advanced age, baseline functional status and the common presence of comorbidities such as chronic heart failure and cognitive impairment among patients with hip fracture are the main arguments in favor of orthogeriatric co-management, which reduces the risk of perioperative complications, functional decline, and mortality [12]. Joint geriatric trauma management was introduced in the United Kingdom in the mid-twentieth century [13]. The past two decades, however, have born an increase in the design and implementation of coordinated perioperative models [14], which have been shown to reduce in-hospital complications [15,16], hospital stay and readmissions [17], disability, and in-hospital mortality [16].

A narrative review published in 2016 [18] considered that geriatric medicine improved awareness of the extra-orthopedic issues complicating the patient’s course and influenced treatment outcomes, improving length of stay, decreasing complication rates and reducing both in-hospital and mid-term mortality after discharge, as well as improving quality of care and reducing healthcare costs. Many questions remain to be answered this field. In addition to the traditional goals of the orthogeriatric team, another crucial objective is enrolling the patient in the most appropriate rehabilitation program, with the aims of reducing institutionalization and facilitating functional recovery and return to the patients’ prefracture social situation [19]. To achieve these goals, correct assessment of the baseline functional situation and maximum potential of recovery are of vital importance. The high prevalence of disability following fracture can condition the patient referral process after hospital discharge [20], and in this sense the management plan does not conclude upon discharge from hospital but rather involves continuation of patient care beyond the in-hospital process. Thus, the scope of the orthogeriatric team goes beyond the hospital setting, expanding the benefits of integral geriatric care [19]. The role of orthogeriatric care has been defined best in the United Kingdom, largely as a consequence of the development of the best practice tariff introduced in 2010 in order to improve the care of patients with hip fracture [21]. Later, preoperative and postsurgical cognitive assessments were also included [22]. The National Institute for Health and Care Excellence issued a document on the quality care of patients with hip fracture that highlighted a number of quality standards to be met in order to maximise efficiency in the care of patients with hip fracture [23]. All orthogeriatric care models have in common the suitability of care provided by multidisciplinary teams proficient in geriatrics, the need of early surgery, the role of a case manager (in this case a geriatrician) throughout the whole process, pain control, avoidance of the appearance or worsening of geriatric syndromes, and adequate continuity of care after hospital discharge, with the aim of recovering baseline function [24]. Orthogeriatric care has been validated by a meta-analysis [8]. However, there are still areas of controversy in need of study and analysis, such as the ideal thromboprophylactic, anaesthetic and analgesic protocols, the assessment and management of cognitive impairment and malnutrition during acute hospitalisation, improvement of patient mobility, postoperative rehabilitation and the efficiency of the programs used in convalescence units or in home rehabilitation care [18,24]. Orthogeriatric co-management exists in several forms, with various types of structural collaboration between orthopedic trauma surgeons and multi-professional geriatric teams. The models are country-specific and there are still no clear recommendations on how this service should be best organized; further studies are needed to determine the best model and to define a uniform set of outcome parameters for use in future clinical studies [25].

The present review aims to provide answers to some of these questions regarding orthogeriatric management of patients with hip fracture, with the goal of clarifying which measures have improved outcomes.

## 2. Methods

The present review was carried out by conducting an electronic search in OVID (MedLine and Embase), combining the following MeSH keywords: “hip fractures” and “geriatric assessment”, combined with “perioperative management” “surgery”, “second hip fracture” and “orthogeriatric care”. The MeSH construction [Hip fractures] AND ([Geriatric assessment] OR “perioperative management”) OR “orthogeriatric care” OR “geriatric syndromes”) was used. The search was limited to publications in the last 5 years; in English, Spanish, and French; and in human subjects.

A total of 783 articles were obtained, of which 124 were finally selected. Some additional instructions were added for certain specific objectives where necessary. In 9 cases, supplementary information was obtained in the form of references of the selected articles. Details of the evaluation and selection process of the items are shown in Figure 1. The articles were selected by four researchers based on the following inclusion criteria: (1) study type: randomized clinical trials, cohort studies, case–control studies, observational studies, and before–after analyses in orthogeriatric units; (2) population: geriatric patients with proximal femoral fracture; (3) intervention: orthogeriatric treatment initiated peri-operatively; and (4) outcomes: surgical delay (defining delayed surgery as that occurring beyond the day after admission, as recommended by the UK’s National Institute for Health and Care Excellence (NICE) clinical guideline for the management of hip fractures [23]), length of hospital stay, prognostic factors and mortality, functional recovery, geriatric syndromes, perioperative care such as renal function, anemia, and costs. The exclusion criteria were letters to the Editor, case reports, articles with no available abstract or those with only the abstract published, and studies meeting the inclusion criteria but with 50% of the study sample aged less than 65 years (i.e., predominantly non-geriatric). All selected studies were included in a database including an abstract of the main results reported. The authors of the review reevaluated all articles, and final inclusion was restricted to those of sufficient quality to provide information pertinent to the aims of this review. In case of discrepancy between the four authors, the fifth author determined including the study or not. The outcome measures examined were mortality, length of hospital stay, medical complications, discharge destination, functional status and recovery. The authors evaluated the different studies according the 12 outcomes parameters proposed by an Expert Roundtable [26].

## 3. Results

A total number of 133 studies we included in this review. Aspects such as race or sex-related differences in the outcomes were only taken into account when reported in the studies. A recent pre-post intervention observational study compared a retrospective control arm (Usual orthopedic care (UOC)) to two parallel arms, orthogeriatric co-management (OGC) and orthopedic team with the support of a geriatric consultant service (GCS). Patients in the OGC group were more likely to undergo surgery within 48 h after admission (OR 2.62; 95% Confidence Interval (CI): 1.40–4.91), but not those in the GCS group, compared with those who received UOC. OGC was also associated with lower length of stay (LOS) and 1-year mortality [27]. In spite of the available evidence, many hospitals still lack this model of care. Another important issue is the collection of data via national registries allowing for audit and comparison of outcomes between the traditional approach and orthogeriatric management, as well between the models available in different countries, in order to define the benefits of the different implemented models [28]. All variants agree in the need for early geriatric clinical care and early surgical management. A 24-h delay may be a threshold for an increased risk of complications and mortality. In a population-based, retrospective cohort study among 42,230 patients with hip fracture, patients who received surgery after 24 h had a significantly higher risk of 30-day mortality (% absolute RD, 0.79; 95%CI, 0.23–1.35) and the composite outcome of complications (myocardial infarction, deep vein thrombosis, pulmonary embolism, and pneumonia) (% absolute RD, 2.16; 95%CI, 1.43–2.89) [29]. Clinical stabilization of patients by the orthogeriatric teams, based on clinical recommendations and guidelines, can help reduce delays, increasing diagnostic precision and risk assessment of comorbidities. Thus, the role of an orthogeriatrician in an orthopaedic department who leads a multidisciplinary approach in the management of older patients with hip fractures is vital, ensuring that surgical delay is under 48 h after presentation, as well as reducing postoperative and total length of stay [30].

The orthogeriatric approach uses an important tool: The Comprehensive Geriatric Assessment (CGA). Two recent meta-analyses on the advantages of this tool in hip fractures patients showed a decrease in mortality and an improvement in activities of daily living [31], physical function and quality of life [32]. By adequately estimating perioperative risk, preventing complications and avoiding heterogeneity in the fulfilment of the goals of care; CGA leads to an important decrease of hospital stay and complications, and prioritizes the recovery of baseline functional and social status. The good results shown are made possible by a continuous improvement in quality of care, reduction of the length of stay in the emergency department, promotion of structured management, and inclusion of new evidence-based measures. Throughout this review, the authors will describe the newest evidence regarding the management of geriatric patients admitted for hip fractures.

### 3.1. Geriatric Syndromes

#### Delirium

The incidence of delirium among orthopedic surgery patients has been reported to be between 4.5 and 41.2%, according to a recent meta-analysis [33]; this wide range in the incidence reported is due to the different ages of the patients included in the studies, the screening tools employed, the types of settings and the surgical and anesthetic techniques used. Risk factors for delirium included advanced age, male sex, comorbidities, malnutrition, preoperative and postoperative haemoglobin levels, postoperative sodium levels, postoperative length of stay, hearing impairment, polypharmacy, antipsychotic drugs, opioid prescription, and cognitive impairment [33].

Analogously, four observational studies have shown an incidence of delirium of 15.7% [34], 22,09% [35], 24.2% [36] and 31.1% [37]. Poeran et al. associated postoperative delirium with long-acting and combined short and long-acting benzodiazepines and ketamine while neuraxial anaesthesia and opioid use were associated with lower risk [34]. Tao et al. identified baseline Barthel index, Mini-Mental State Examination (MMSE), instrumental activities of daily living (IADLs), and Geriatric Depression Scale (GDS) as risk factors for delirium [35]. Aldwikat et al. found that comorbidity and cognitive impairment were independent risk factors for the development of delirium [36]. Finally, Pioli et al. linked risk of delirium to older age, higher degree of comorbidity and functional impairment [37]. In addition, in the multivariate analysis, surgical delay was an independent risk factor for delirium, along with age, prefracture functional disability and cognitive impairment (mildly to moderately impaired versus cognitively healthy or severely impaired patients). All the described factors were included in the previously cited meta-analysis [33].

Another two meta-analyses identified risk factors already described in the one mentioned above. Yang et al. reported an incidence of delirium of 24.0 % and found preoperative cognitive impairment, advanced age, living in an institution, heart failure, total hip arthroplasty, multiple comorbidities and opioid usage as risk factors for delirium, while females were less likely to develop delirium after hip surgery [38]. Smith et al. found age 80 years and over, living in a care institution before the fracture and the pre-admission diagnosis of dementia to be factors associated with the appearance of delirium, which occurred in 31.2% [39].

Delirium is associated with increased mortality, with the aforementioned confounders responsible for the statistically significant association between incident postoperative delirium and mortality, as shown in a meta-analysis published in 2017 [40].

There is an overlap between the different geriatric syndromes, emphasizing the need of CGA in older patients admitted by hip fracture. Patients that were malnourished (OR = 2.98; 95%CI: 1.43–6.19), or at risk of malnutrition (OR = 2.42; 95%CI: 1.29–4.53) had an increased risk of delirium [41]. Other known risk factors for delirium are cognitive impairment and dementia. An observational study reported that 52% of patients developed delirium in addition to dementia, and as this overlap increased length of stay and short-term mortality [42]. A study published before the period included in this review already showed an overlap of geriatric syndromes between depressive symptoms and incidence of delirium (21.7% of patients); other syndromes overlapping with delirium in this study were vision impairment and lower cognitive function [43].

It is perhaps due to this overlap of geriatric syndromes that strategies for prevention and treatment of delirium based on proactive geriatric consultation have shown a decrease in the incidence of delirium. Two meta-analyses published in 2017 analysed the role of CGA in reducing the incidence of delirium. The first one found a significant reduction in delirium overall (relative risk [RR] = 0.81; 95% = 0.69–0.94) in the intervention group. Post-hoc subgroup analysis found this effect to be maintained in the team-based intervention group (RR = 0.77; 95%CI = 0.61–0.98) but not the ward-based group [44]. The second one showed that comprehensive geriatric care reduced the incidence of perioperative delirium (Odds ratio [OR] = 0.71; 95%CI = 0.57–0.89; *p* = 0.003) and that it was associated with better cognitive status during hospitalization or at 1-month follow-up (MD = 1.03; 95%CI, 0.93–1.13; *p* ≤ 0.00001) but there was no significant difference in the duration of perioperative delirium between both groups (MD = −2.48; 95%CI = −7.36–2.40; *p* = 0.32) [45]. Most of the risk factors described for delirium are evaluated during the CGA process, and this assessment reduces delirium incidence by identifying potential risk factors and developing preventive strategies.

Another important strategy to decrease the incidence of delirium incidence is prompt surgery. Surgical delay is linked to delirium, as has been previously mentioned [37], but the duration of surgery is associated to delirium risk, too. A recent study reported that the risk for delirium was increased with surgical duration: every 30-min increase in the duration of surgery was associated with a 6% increase in the risk of delirium, and the risk was higher was in patients who had received general anaesthesia [46].

### 3.2. Cognitive Impairment

Both cognitive impairment and dementia are quite prevalent among patients with hip fractures. Furthermore, rather than maintaining their cognitive function, older patients with cognitive decline could further develop cognitive disorders after hospitalization following hip fracture. An observational study included 402 patients with hip fracture, 188 of whom were previously cognitively intact. Of these, 12 (6.4%) patients showed a cognitive decline in the 6 months following the fracture. Multivariate regression analysis showed that older age and the appearance of medical complications were significant independent risk factors for cognitive decline [47]. Older studies have reported from a 40% prevalence of some degree of cognitive impairment [48] to an 85% prevalence of dementia [49]. Dementia is underdiagnosed in older hospitalised patients, in spite of being an important risk factor for suffering hip fractures [49].

Detecting cognitive impairment is vital, as one of the most important risk factors of functional decline and mortality in hip fracture patients is concomitant dementia. A recent observational study found that patients with cognitive impairment showed a higher overall mortality, even after discharge from hospital [50]. This study reported that patients with dementia were more likely to suffer respiratory infections, urinary tract infections, sepsis, had a poorer baseline functional status, and worse ambulation at final follow-up [50]. The relationship between function and cognitive decline is well known [51]. Only 31% percent of hip fracture patients recovered previous activities of daily living (ADL) ability in an observational study, and recovery was less likely for those ≥85 years old, with dementia and with a Charlson Comorbidity Index > 2 [52]. Another study reported that hip fracture patients with worse scores in a cognition scale had less functional gain, but those who improved their cognitive score showed better recovery of gait ability [53], demonstrating the benefits of a dual cognitive and functional rehabilitation in these patients. For this reason, rehabilitation protocols for functional recovery in hip fracture patients with cognitive impairment or dementia should include cognitive stimulation programs, as well. As previously mentioned, the negative association between dementia in hip fracture patients and mortality can be attenuated by geriatric care. A study of 650 patients (mean age 86 years (standard deviation [SD]: 6 years)) identified 168 patients with dementia (DP), 400 patients without dementia (NDP) and 82 patients with in whom cognitive status was not determined (CSND). After adjusting for age, sex, comorbidities, polypharmacy, pre-fracture independence, time-to-surgery, and delirium, there were no significant differences between groups for mortality or for recovery of ambulation at 6 months, but DP and CSND were more likely to be newly institutionalized, being possible to attribute this absence of difference to the effect of a dedicated geriatric care pathway [54]. Thus, treatment of behavioral psychological symptoms of dementia (BPSD) during rehabilitation is crucial. A retrospective cohort study based on the Japan Rehabilitation Database showed that participants who presented BPSD when initiating rehabilitation but who had resolved their symptoms at the end of rehabilitation had better functional recovery [55]. Likewise, the goals of rehabilitation in hip fracture patients with dementia should focus not only on functional recovery, but rather add other objectives such as quality of life, decrease in the complication rate or optimization of social support [56].

Finally, a study among older veterans diagnosed with dementia suggested that acetylcholinesterase inhibitors (AChEI) could reduce the risk of fragility fractures by increasing bone density and quality, as well as improving bone healing after fracture [57]. Over an average of 4.6 years of follow-up, 20.1% suffered a fracture, and 42.3% of the cohort had been prescribed AChEIs. The hazard of suffering any fracture was significantly lower among AChEI users compared with those on other/no dementia medications in fully adjusted models (hazard ratio [HR] = 0.81; 95%CI: 0.75–0.88). After considering competing mortality risks, fracture risk remained 18% lower in veterans using AChEIs (HR = 0.82; 95%CI: 0.76–0.89) [57]. This is a field of interest for research to confirm if AChEI would be useful to prevent hip fracture or increase bone healing after surgery in patients with dementia. 

### 3.3. Mood Disorders and Depression

The prevalence of mood disorders is high among hip fracture patients and depression and its treatment increase the risk of fractures and have a negative impact on functional recovery and mortality [58]. Van de Ree et al. reported a prevalence of psychological distress of 36% at 1 week to 31% at 1 year after hip fracture. Frailty at presentation for hip fracture was the most important prognostic factor for symptoms of depression (OR = 2.74; 95%CI = 1.41–5.34) and anxiety (OR = 2.60; 95%CI = 1.15–5.85) in the year following hip fracture [59]. Again, overlapping of geriatric syndromes overlap has important consequences for older patients with hip fractures and highlight CGA-based intervention strategies that involve early identification of geriatric syndromes and provision of appropriate and prompt treatments.

Depression is common among hip fracture elderly patients: a cross-sectional study reported and overall prevalence of 46%, significantly higher in women, persons over 81 years old, diabetics and those with anxiety [60]. These studies provide further proof of the need of routine geriatric assessment in older patients hospitalized after hip fracture. A secondary analysis of data from a randomized controlled trial comparing usual care with an interdisciplinary program evaluated differences in depressive symptoms using the Chinese version of the Geriatric Depression Scale short-form. Three trajectory groups were defined according to changes in depressive symptoms: a non-depressive group, a marginally depressive group and a persistently depressive group. Compared to patients who received usual care, those in the interdisciplinary program had a significantly lower risk of being in the persistently depressive group [61]. Women and those physically and cognitively more impaired were found to be more likely to be assigned to the marginally and persistently depressive groups. Screening of depression could contribute to managing it better and minimizing its negative impact on patient recovery.

#### 3.3.1. Urinary Incontinence

Urinary incontinence (UI) is another highly prevalent geriatric syndrome among older patients with hip fracture. In a randomized clinical trial, 44% of study participants self-reported UI and four out of five reported nocturia at baseline [62]. A cohort study demonstrated that UI was associated to an increased risk of falls, but not of hip fractures [63]. Post-void residual (PVR) urine volume was elevated in 15.6% of patients included in a prospective observational study, and elevated PVR was more likely in the setting of urinary or fecal incontinence, difficulties in activities of daily living, malnutrition, poor performance on Timed Up and Go test and Elderly Mobility Scale. One-year mortality after hip fracture was significantly higher among those with elevated PVR. PVR deserves to be included in the CGA of frail older patients, including women [64]. Post-operative urinary retention (POUR) is common after hip fracture surgery, and is linked to opioid use and anticholinergic medication. The high incidence of asymptomatic POUR in older patients underscores the need of improved screening tools for early identification and treatment of this condition [65]. Half of the population was unable to recover their prefracture autonomy in a prospective cohort study. Risk factors for not recovering autonomy were increasing age, number of comorbidities, lower prefracture autonomy, increased use of an anti-decubitus mattress, more days with diapers, a urinary catheter or bed rails, a higher number of days with disorientation, failure to recover ambulation, and a nonintensive care pathway. Recovery of ambulation, treatment of disorientation and management of urinary incontinence are modifiable factors significantly associated with the functional recovery of autonomy [66]. Health professionals should be aware of the high prevalence of urinary problems in older adults with hip fractures, and screening tools and early management should be implemented in these patients.

#### 3.3.2. Constipation

Constipation is also common among patients admitted due to hip fracture. I has been reported to be associated with immobilization, loss of intimacy, polypharmacy and treatments such as opioids. Approximately 70% of all patients develop constipation the first days after surgery, and 62% continue to suffer from it 1 month after surgery [67]. Some multicomponent interventions included in the CGA could reduce the incidence of constipation. A quasi-experimental study testing the efficacy of a nursing intervention based on active patient involvement including individualised nursing care plans reported significant lower rates of constipation in the intervention group, attributed to higher fibre and fluid intakes [68].

## 4. Malnutrition

Nutritional problems have a reported prevalence between 9 and 18,7% among older patients hospitalised due to hip fracture according to recent studies, and 50% of patients are at risk of malnutrition.

In a retrospective cohort study of 29,377 geriatric patients 45.9% had hypoalbuminemia, and the risk of mortality was inversely associated with serum albumin concentration as a continuous variable. Compared with normoalbuminemic patients, hypoalbuminemic patients had higher rates of death, sepsis and unplanned intubation, as well as a longer length of stay. Hypoalbuminemia is a powerful independent risk factor for mortality [69]. A systematic review including 44 articles and 26,281 subjects found a prevalence of malnutrition of 18.7% (using the Mini-Nutritional Assessment (MNA) (large or short form)) that increased to 45.7% when different criteria were used (such as Body Mass Index (BMI), weight loss, or albumin concentration). Low scores in anthropometric indices were associated with a higher risk of in-hospital complications and with poorer functional recovery. Despite improvement in the management of geriatric patients with hip fractures, mortality remained unacceptably high (30% at 1 year and up to 40% at 3 years) and malnutrition was associated with a higher risk of dying [70]. Nutritional assessment as part of the CGA including nutritional screening tools and serum parameters such as albumin is cost effective, improves nutritional status and functional recovery. At baseline, 9% patients were malnourished and 42% patients at risk of malnutrition among 472 hip fracture patients aged 65 years and older included in a population-based prospective study that used baseline Mini-Nutritional Assessment Short Form (MNA-SF) scores. Malnutrition was associated with mortality. Risk of malnutrition and malnutrition also predicted institutionalization, and the risk of malnutrition was associated with decline in mobility in the multivariate binary logistic regression analyses [71]. In a prospective study that included 509 patients (mean age 85.6 (SD 6.9) years, 79.2% female), 20.1% had a BMI lower than 22 kg/m^2^; 81.2% had protein and 17.1% had both energy and protein malnutrition. Serum vitamin D was <30 ng/mL in 93% of patients and 17.1% were sarcopenic. There is an overlap between protein and energy malnutrition, vitamin D deficiency and sarcopenia [72].

Nutritional impairments, vitamin D deficiency and sarcopenia have been associated with functional decline, length of stay, complications such as sepsis and mortality. Furthermore, nutritional assessment has been reported to be cost-effective, and should be included in routine CGA in elderly patients admitted for hip fracture. The question regarding which is the best nutritional screening tool remains open. Three studies evaluated which best tool was best to diagnose malnutrition. Helminen et al. performed a prospective study in which 7% of patients were malnourished and 41% at risk of malnutrition at baseline, according to the MNA-SF. The MNA-SF predicted mortality, LOS and readmissions better than the NRS2002 (Nutritional Risk Score 2002), while both were ineffective in predicting changes in mobility and living arrangements [73]. Inoue et al. compared the MNA-SF, MUST (Malnutrition Universal Screening Tool), NRS-2002 and GNRI (Geriatric Nutritional Risk Index) in 205 patients. Multiple linear regression revealed that MNA-SF was associated with the motor-FIM (functional independence measure) at discharge, efficiency on the motor-FIM relative to length of stay and 10-m walking speed. The GNRI was associated with 10-m walking speed, but not motor-FIM or motor-FIM efficiency. MNA-SF was identified as the ideal nutritional screening tool predict functional outcomes during the acute postoperative phase in older hip fracture patients [74]. Finally, Koren-Hakim compared MNA-SF, MUST and NRS-2002 in 215 patients (71.6% female; mean age 83.5 (SD 6.09)) and found a.significant relationship between the nutritional groups of the three scores. For all screening tools, body mass index, weight loss and pre-admission food intake were related to the patients’ nutritional status. Only the MNA-SF was able to detect the well-nourished patients that would have less readmissions in the 6 months after the fracture. Well-nourished patients according to the MNA-SF had lower mortality at 36 months than malnourished patients and those at risk of malnutrition. The association between the NRS-2002 patients’ nutritional status and mortality was weaker [75]. According to these studies, the MNA could be the best nutritional screening tool for hip fracture patients and would offer the best prediction of survival and functional recovery.

Several studies have evaluated functional recovery among patients with nutritional impairments after hospital discharge. A retrospective observational cohort study divided patients into two groups based on MNA-SF scores at discharge vs. admission: improvement in nutritional status (IN group) and non-improvement in nutritional status (NN group). Patients in the IN group were younger and had higher admission FIM and MNA-SF scores. The median FIM score at discharge was significantly higher in the IN group than in the NN group. Multivariate analysis revealed a significant association between improvement in nutritional status and higher FIM scores at discharge [76]. Another retrospective cohort study analysing 107 rehabilitation patients aged ≥65 years and older reported that compared to lower-functioning patients, higher-functioning patients were younger, were hospitalised less time, and had lower Cumulative Illness Rating-Scale for Geriatrics (CIRS-G) scores with higher mean Mini-Mental Status Examination (MMSE) scores. The gain in FIM was significantly higher in patients at low risk of malnutrition (according to the Short Nutritional Assessment Questionnaire, SNAQ), in those who did not lose weight, had normal albumin, and lower CIES-G scores. Patients who achieved functional independence–discharge FIM ≥ 90–ate normally and experienced less “loss of appetite”. Weight loss was the strongest negative predictor of the gain in FIM. Nutritional status, especially weight change, is an independent negative predictor for the success of rehabilitation [77]. A multicenter prospective cohort study evaluated nutritional status using the MNA-SF in 204 patients: 51 (25.0%) patients were malnourished, 98 (48.0%) were at risk of malnutrition, and 55 (27.0%) were well-nourished before the fracture. At discharge, FIM scores were higher in well-nourished patients than in those malnourished or at risk of malnutrition (*p* < 0.01). MNA-SF remained a significant independent predictor for FIM at discharge even after adjusted multiple regression. The baseline nutritional status was a significant independent predictor for functional status at discharge from acute admission [78]. Finally, a prospective observational cohort study of 254 geriatric patients undergoing surgery showed that most followed one of the five trajectories at one-year: (1) 30% (*n* = 63) returned home, (2) 11% (*n* = 22) returned to a nursing home, (3) 16% (*n* = 36) needed rehabilitation, (4) 13% (*n* = 28) were discharged to a location different from that prior to admission and (5) 18% (*n* = 37) had died. Patients following trajectory 1 were younger while those in trajectory 5 had lower MNA scores. Delay between discharge from the attending staff and true departure from the hospital was associated with low MNA scores, low MMSE scores and with the need for a rehabilitation centre (trajectory 3) [79]. Early assessment of nutritional status and early intervention are important for successful postoperative rehabilitation.

A subanalysis of a randomized controlled trial of orthogeriatric care included nutritional advice and supplementation in the intervention group (orthogeriatric care). Vitamin K1 and 25-(OH)-D levels were higher at 4 months in the intervention group than in controls. No difference was found in bone turnover markers between groups, but a substantial loss of weight and physical function was found in both groups [80].

### 4.1. Sarcopenia

Sarcopenia is partially dependent of nutritional status. The following risk factors of sarcopenia were identified in a multicenter prospective observational study: undernutrition (body mass index-BMI and Mini Nutritional Assessment-Short Form or MNA-SF), hand-grip strength and skeletal muscle index. During follow-up, 114 patients died (60.5% sarcopenic vs. 39.5% non-sarcopenic, *p* = 0.001). Cox regression analyses showed that sarcopenia and low hand-grip strength were associated with an increased risk of dying. Older patients with undernutrition had a higher risk of developing sarcopenia during hospitalisation, and sarcopenic patients were almost twice as likely to die during follow-up after hip fracture [81]. Using the European Working Group on Sarcopenia in Older People Criteria (EWGSOP), a prospective study of 479 consecutive patients hospitalized for hip fracture identified sarcopenia in 17.1%. Sarcopenia was associated with living in nursing homes, older age, and having a lower body mass index, but only low body mass index was predictive of sarcopenia after adjustment in the multivariate analysis [82]. A third study assessed sarcopenia using the SARC-F self-reported questionnaire and found a prevalence of 63.5%. The sensitivity, specificity, positive predictive value, negative predictive value were 95.35 %, 56.94 %, 56.94%, 95.35%, and 71.3%, respectively versus the EWGSOP-2 criteria as the reference standard [83], suggesting SARC-F could be useful to identify sarcopenia in hip fracture patients. This is particularly important as sarcopenia is linked to poorer functional recovery. Another study diagnosed sarcopenia using the definition of the Foundation for National Institutes of Health (FNIH) criteria in 127 patients (mean age of 81.3 (SD 4.8) years, and 64.8% female) and identified sarcopenia in 33.9%. Participants with sarcopenia were less likely to have complete functional recovery and showed lower Barthel index scores at discharge from the rehabilitation unit [84].

In summary, sarcopenia is common among older patients with hip fractures, and is associated with a poorer nutritional status and lower likelihood of functional recovery in rehabilitation programs. These patients could benefit of the development of personalized treatment plans that include nutritional and functional interventions.

### 4.2. Frailty

Frailty is another geriatric syndrome highly prevalent in older patients with hip fracture and has been associated with the incidence of complications and length of stay. Of 696 patients aged 65 years and older included in a prospective cohort study, 53.3% were considered frail. Frailty was negatively associated with health status, self-rated health and capability well-being 1 year after hip fracture, even after adjusting for confounders [85]. Another study evaluated the value of the ASA (American Society of Anaesthesiologists) score and Edmonton frailty score in predicting the outcome of treatment of femoral neck fractures in older patients. The frailty index, calculated using Edmonton scoring index, showed that 49% had low frailty scores and 51% had high frailty scores. Patients with high frailty scores and ASA grade had a greater chance of developing wound infection, as well as higher morbidity and mortality following femoral neck fracture [86]. Frail patients had a significantly lower survival compared to nonfrail patients in a prospective observational cohort study [87]. The final study included in this subsection used the 7-point Clinical Frailty Scale to diagnosis frailty in 164 patients: 81 patients were ‘not vulnerable’ (frailty score 1–3) and 83 were ‘vulnerable or frail’ (frailty score ≥ 4). One month after surgery, 5% patients had died, all of them with frailty scores ≥ 4 (*p* = 0.007). Postoperative morbidity during the 28-day follow-up was less common among patients categorised as ‘not vulnerable’. Postoperative length of stay was longer for ‘vulnerable or frail’ with scores ≥ 4 [88]. Frail patients also show a lower chance of functional recovery: in a study of 100 consecutive hip fracture patients (mean age 79.1 (SD 9.6) years), 37.8% had post-operative complications. Frailty, measured using the MFC (modified fried criteria) and REFS (reported Edmonton frail scale), was significantly associated with suffering complications using both scales (OR = 4.46, *p* = 0.04 and OR = 6.76, *p* = 0.01, respectively), which were the only significant predictors of post-operative complications on univariate analyses. However, only REFS (OR = 3.42, *p* = 0.04) predicted early post-operative complications in the hierarchical logistic regression model. REFS also significantly predicted [basic activities of daily living (BADL)] function at 6-month follow-up in the multivariable logistic regression models. (BADL, OR = 6.19, *p* = 0.01). Frailty, measured with the REFS, was a good predictor of early post-operative outcomes in this pilot study of older adults undergoing hip surgery, and it also predicted 6-month BADL function [89].

### 4.3. Pressure Sores

Pressure sores are a geriatric syndrome commonly presenting during hospitalisation after hip fracture. Proof of their importance is that national hip fracture audits include pressure sores as a variable. A study comparing the results reported by different national hip fracture registries described an incidence of pressure sores between 2 and 6.7% [28]. These rates are lower than those described in cohort studies and meta-analysis, as we shall discuss later. The difference can be possibly explained due to the fact that registries are based on health records and rely of the quality of this clinical information.

Pressure sores are more common in some diseases such as diabetes. A meta-analysis reported that 15.1% of diabetics had pressure sores, compared to 7.5% among hip fracture patients without diabetes. The risk of pressure ulcers during hospitalisation was increased in diabetics with hip fractures (OR = 1.825 [95%CI: 1.373–2.425) [90]. Geriatric care needs to intensify preventive measures in these patients. Pressure ulcers are also associated with surgical delay: a meta-analysis showed an increase in complications including pressure ulcers among patients with higher surgical delay [91].

Approximately 12% of patients suffered category II or higher pressure ulcers in a prospective cohort study that identified five risk factors associated with developing sores: higher preoperative Braden score, surgical procedure with internal fixation, a higher percentage of days with the presence of foam valves before surgery, use of a urinary catheter, and use of a diaper in the postoperative period [92]. Another prospective cohort study also found an incidence of 12% and linked this geriatric syndrome to low albumin levels, history of atrial fibrillation, coronary artery disease and diabetes. Pressure ulcers were also associated with 6-month mortality (RR = 2.38, 95%CI = 1.31–4.32, *p* = 0.044) [93].

In another cohort study of 8871 geriatric hip fracture patients, 457 (5.15%) developed pressure ulcers. Risk factors of developing pressure ulcers were preoperative sepsis, elevated platelet count, insulin-dependent diabetes, pre-existing pressure ulcers, postoperative pneumonia, urinary tract infection, and delirium [94]. Pressure sores appeared in 22.7% of 1083 older adult patients with fragility hip fractures included in a prospective multicentric prognostic cohort study; risk factors identified were: age over 80 years, the length of time an indwelling urinary catheter was used, duration of pain, the absence of side rails on the bed, and the use of a foam position valve [95]. The incidence of pressure ulcers was 25.7% in a cohort study of 462 patients with hip fracture. The incidence was higher in weaker subjects, and baseline Barthel index, and MNA scores were lower among those developing ulcers. However, only low handgrip strength remained associated with the development of pressure ulcers upon multivariate adjustment [96].

The effects of multidisciplinary co-management of older hip fracture patients were evaluated in a retrospective study that included 3540 patients. Half of the patients who received co-management received surgery within 48 h of ward admission, compared to 6.4% before the intervention, 0.3% (vs 1.4%) developed pressure ulcers, and 76% (vs 19%) were assessed for osteoporosis [97].

In a prospective prognostic cohort study of patients admitted with fragility hip fractures and monitored over a 12-month period, 27% developed pressure sores. Multivariate analysis identified the following risk factors: age older than 81 years, type of surgery, and placement of the limb in a foam rubber splint. Pressure ulcers are a relatively common complication in older adults with hip fractures, especially high-risk patients or with certain treatments. Pre-emptively identifying patients at highest risk of pressure injury taking these factors into account could help provide and targeted care [98].

## 5. Polypharmacy

Polypharmacy, fall-risk increasing drugs and inadequate prescription are very common in older adults. A retrospective cohort study analysed polypharmacy and fall-risk increasing drugs (FRIDS) in 228 patients older than 80 years discharged from an Orthogeriatric Unit who were able to walk before surgery. The mean number of drugs and FRIDS prescribed at discharge was 11.6 (SD 3.0) and 2.9 (SD 1.6), respectively. Polypharmacy was very prevalent: 23.3% (5–9 drugs) and 75.9% (≥10 drugs); only three patients did not meet the definition of polypharmacy. In addition, only 11 patients had no FRIDS and 35.5% were on <3 FRIDS. The most prevalent FRIDS were: agents acting on the renin-angiotensin system (43.9%) and anxiolytics (39.9%). The number of FRIDS was higher in patients with extreme polypharmacy. Those independent in instrumental activities had lower risk of extreme polypharmacy (≥10 drugs), while patients living in a nursing home had higher risk of >3FRIDS [99].

Orthogeriatric co-management with CGA based care could help stop inappropriate prescriptions. The differences in drugs prescribed at admission and discharge were analysed in a randomized clinical trial that compared comprehensive geriatric care (CGC) in a geriatric ward with traditional orthopaedic care (OC). The mean number of drugs prescribed at discharge in the CGC group was lower compared with OC (7.1 (SD 2.8) versus 6.2 (SD 3.0)) and the total number of withdrawals and of starts was higher in the CGC group. The number of drug changes during hospitalisation was negatively associated with mobility and function at 4-month follow-up in both groups, but this association disappeared in multivariate analysis using baseline function and comorbidities as a confounders [100]. CGA interventions including assessment of drugs prescription at hospital discharge could have a potential impact on adverse events and the incidence of falls in older patients. Table 1 summarizes the most important papers on geriatric syndromes included in this review. The studies were included in this selection according to the level of evidence and the authors’ consideration of their clinical relevance.

## 6. Perioperative Care

### 6.1. Renal Function

Low glomerular filtration rates have been associated with increased comorbidity, lower haemoglobin concentrations at admission, longer surgical delay, and greater incidence of delirium. Of 1425 consecutive hip fracture patients included in a population-based prospective study, 40% had renal dysfunction on admission using the Chronic Kidney Disease Epidemiology equation (eGFRCDK-EPI) [101]. In the multivariate analyses, eGFRCDK-EPI values of 30–44 mL/min/1.73 m^2^ (HR = 1.91; 95%CI = 1.44–2.52) and <30 mL/min/1.73 m^2^ (HR = 1.95; 95%CI = 1.36–2.78) were associated with increased mortality. In summary, moderate to severe renal dysfunction measured by eGFRCDK-EPI and polypharmacy increased mortality after hip fracture. Frequent assessment of renal function and medications are essential in the care of geriatric hip fracture patients.

### 6.2. Anemia and Patients Blood Management

Approximately 40% of all hip fracture patients have haemoglobin (Hb) values below 12 g/dL upon admission to hospital. Anaemia progresses significantly during the days before surgery, more so in extracapsular fractures. In hip fracture patients, anaemia has been associated with increased risk of blood transfusion, poorer functional outcomes and increased mortality [102]. Hip fracture surgery is additionally associated with perioperative blood loss frequently requiring transfusion. Patient blood management (PBM) involves multidisciplinary strategies to optimize outcomes. The management of anaemic patients includes preoperative fluid resuscitation, the administration of iron alone or combined with vitamin B12, folic acid, and on occasion erythropoietin, as well as blood products; it also includes the minimization of further intraoperative and perioperative losses.

Some risk factors for increased hidden blood loss after a hip fracture are higher ASA score, perioperative gastrointestinal bleeding/ulcer and use of general anaesthesia compared to spinal anaesthesia. Patients with higher hidden blood loss were more likely to receive transfusions [103]. Advanced age, preoperative anaemia, female sex, lower BMI, higher ASA scores, chronic obstructive pulmonary disease (COPD), hypertension, increased surgical delay, and having intertrochanteric and subtrochanteric femur fractures were perioperative independent risk factors associated with receiving postoperative blood transfusions in older patients with hip fractures included in the American College of Surgeons National Surgical Quality Improvement Program (ACS NSQIP) [104]. Patients receiving postoperative transfusions had a significantly higher risk-adjusted 30-day mortality, total hospital length of stay and readmission rates. Survival at 90 days, 180 days, and one year after surgery was significantly lower among patients with a Hb level below 12 g/dL at admission [105].

The 2018 PBM International Consensus Conference defined the current status of the PBM evidence base for clinical practice in major orthopaedic surgery. It recommended using intravenous (IV) iron for patients with iron deficiency anaemia to reduce red blood cell (RBC) transfusion rates; erythropoietin therapy in addition to IV iron in patients with Hb levels < 13 g/dL; and it also established a conditional recommendation in favour of using a RBC transfusion threshold of Hb < 8 g/dL in adults with hip fractures and cardiovascular disease or risk factors [106].

PBM-based strategies for the prevention and treatment of anaemia and transfusion have demonstrated an improvement of outcomes after hip fracture. A meta-analysis comparing restrictive versus liberal transfusion strategies in patients undergoing hip fracture surgery found no differences in the rates of delirium, mortality, the overall incidence of infections, the incidence of pneumonia, wound infection, cardiovascular events, congestive heart failure, thromboembolic events or length of hospital stay between restrictive (haemoglobin level threshold ≤8 g/dL or symptoms) and liberal (Hb level threshold ≤10 g/dL) RBC transfusion strategies (*p* > 0.05). However, the authors found that restrictive transfusion thresholds were associated with higher rates of acute coronary syndrome and a 40% decrease in the risk of cerebrovascular accidents. The authors concluded that that clinicians should individualise treatment based on patient condition before adopting a transfusion strategy, rather than using haemoglobin level thresholds [107]. In a retrospective study, a restrictive transfusion strategy was associated with fewer acute cardiovascular complications and a reduction in packed RBC units used per participant, but also with a greater frequency of transfusion in the rehabilitation setting [108]. Another retrospective cohort study compared a restrictive (transfusion threshold of haemoglobin <8 g/dL) with a very restrictive transfusion protocol (threshold of <7 g/dL Hb in hemodynamically stable patients and <8 g/dL in patients with symptomatic anaemia or a history of coronary artery disease); the very restrictive protocol decreased transfusion rates, a lower likelihood of transfusion of more than 1 unit of RBCs, and lower inpatient cardiac morbidity without differences in morbidity, in-hospital mortality and readmission and survival at one month follow-up [109].

Intravenous iron is an alternative to avoid RBC transfusion. A meta-analysis comparing iron supplementation with placebo in 1201 patients undergoing hip fracture surgery, found that administering 200–300 mg iron IV preoperatively was associated with a reduction in transfusion volume and length of stay, but was not found to reduce infections or mortality [110]. Preoperative iron supplementation combined with restrictive transfusion strategy (Hb level threshold ≤ 8 g/dL or symptoms) was compared with a liberal transfusion strategy (Hb level threshold ≤ 10 g/dL) without iron supplementation during hospitalization for hip fracture in a retrospective cohort study. The restrictive transfusion strategy was associated with a reduction in packed RBC units used per patient, but more transfusions in rehabilitation settings [111].

The combined use of IV iron and erythropoietin (EPO) did not reduce the percentage of transfused patients in two cohort studies [112,113] but it did reduce the number of RBC units required. Patients in the intervention group showed improved functional recovery at 3 and 6 months after the fracture, measured with the Barthel index and the Functional Ambulation Categories (FAC scale) [112]. A retrospective study compared RBC transfusion with a patients treated with iron and EPO [114]. The transfusion group had higher haemoglobin levels on the first postoperative day without differences in mortality; haemoglobin levels were completely recovered within 2 weeks in both groups. Treatment with EPO could improve functional recovery as well, as suggested by a randomized clinical trial [115] that used EPO in sarcopenic patients with femoral intertrochanteric fractures and reported a higher handgrip strength in sarcopenic women in the intervention group, but not in men. The appendicular skeletal muscle increment of the intervention group was markedly increased regardless of sex. The postoperative infection rate and length of stay were lower in the intervention group. In summary, EPO could improve the muscle strength of female patients with sarcopenia during the perioperative period-but not revert sarcopenia itself. EPO could also increase muscle mass in both sexes. Postoperative administration of EPO could therefore potentially accelerate postoperative rehabilitation.

Intravenous tranexamic acid (TXA) is another option in PBM. It possesses great potential in reducing blood loss and allogeneic blood transfusion safely in patients with hip fractures undergoing surgery. Five meta-analyses [116,117,118,119,120] of RCTs comparing intraoperative administration of TXA with placebo in patients undergoing hip fracture surgery showed significant differences between groups regarding transfusion rates of allogeneic blood, total blood loss, intraoperative blood loss, postoperative blood loss and postoperative haemoglobin, without affecting the rates of thromboembolic events, deep venous thrombosis, acute coronary syndrome, cerebrovascular events, wound complications or mortality.

### 6.3. Pain Management

Insufficient control of pain during hospitalisation for hip fracture has been associated with an increased incidence of delirium and poorer outcomes. A review published in 2016 warned of the importance of pain associated with hip fracture due to its severe consequences and delayed recovery. However, the prevailing opioid-dependent model of analgesia, presents multiple disadvantages and risks that can compromise outcomes in the hip fracture population. The pain management process includes fundamental preoperative, intraoperative, and postoperative interventions and lacks sufficient well-designed studies to unequivocally show which pain management approaches work best after hip fracture surgery [121].

A study used the initial pain evaluation by emergency medical services using the Numeric Rating Scale (NRS) and reported that 28% of patients received analgesics, with their score dropping from 7.0 (SD, 2.6) to 2.8 (SD, 1.4) upon hospital arrival [122]. The authors of this study highlighted that only a minority of patients received pre-hospital analgesia and this treatment was linked to significant pain relief. Treatment of pain during transfer to hospital could be implemented in hip fracture treatment guidelines.

Pain was measured with the Western Ontario and McMaster Universities Osteoarthritis Index questionnaire’s short form (WOMAC-SF) in a prospective study [123]. Predictors of worse pain at six or eighteen months after the fracture were: living in a home care situation or nursing home before the fracture and low pre-fracture pain. Predictors of functional deterioration at six months were: age ≥85 years, lower income, high pre-fracture hip function, referral to rehabilitation upon discharge, and longer surgical delay. In summary, prefracture frailty is a predictor of greater post-fracture pain and functional decline. Prevention of frailty by promoting exercise in older adults could improve the prognosis following hip fracture.

The application of femoral nerve blocks in the Emergency Department among older adults with acute hip fracture has been evaluated in a systematic review that included seven randomized controlled trials [124]. All reported reductions in pain intensity with femoral nerve blocks, and all studies but one reported a decrease in the requirements of rescue analgesia. No adverse effects were found to be associated with the femoral block procedure; in fact, two studies reported a decreased risk of adverse events such as respiratory and cardiac complications. Femoral nerve blocks are beneficial both in terms of decreasing the pain experienced by older patients, as well as limiting the amount of systemic opioids administered. A Cochrane systematic review and meta-analysis on peripheral nerve blocks (PNBs) for hip fractures in adults included 49 trials (3061 participants; 1553 randomized to PNBs and 1508 to no nerve block (or sham block)) published from 1981 to 2020 [125]. The average age of participants ranged from 59 to 89 years. People with dementia were often excluded from the included trials. The results of 11 trials with 503 participants showed that PNBs reduced pain on movement within 30 min of block placement (standardized mean difference (SMD) −1.05, 95% confidence interval (CI) −1.25 to −0.86; equivalent to −2.5 on a scale from 0 to 10; high-certainty evidence). The effect size was proportional to the concentration of local anaesthetic used (*p* = 0.0003). Based on 13 trials with 1072 participants, PNBs decreased the risk of acute confusional state (RR = 0.67; 95%CI = 0.50–0.90; number needed to treat for an additional beneficial outcome (NNTB) = 12, 95%CI 7–47; high-certainty evidence). PNBs are likely to reduce the risk for chest infection (RR = 0.41 95%CI = 0.19–0.89; NNTB = 7, 95%CI 5–72; moderate-certainty evidence). The effects of PNBs on six-month mortality are uncertain, due to very serious imprecision (RR = 0.87, 95%CI = 0.47–1.60; low-certainty evidence). PNBs are likely to reduce time to first mobilization (mean difference (MD) −10.80 h, 95%CI: −12.83 to −8.77 h; moderate-certainty evidence). In summary, PNBs reduce pain on movement within 30 min after block placement, risk of acute confusional state, and probably also reduce the risk of chest infection and time to first mobilization.

A randomized clinical trial examined the effect on pain intensity and mobility of incorporating transcutaneous electrical nerve stimulation (TENS) treatment added to standard rehabilitation care during the acute post-operative phase following Gamma-nail surgical fixation of extracapsular hip fractures. The authors reported a significantly greater pain reduction during walking in the active TENS group compared to sham TENS group. Additional improvements in the active TENS group were a greater increase in walking distance on the fifth postoperative day and a higher level of mobility compared to the sham TENS group. The authors concluded that adding TENS to the standard care of elderly patients in the early postoperative period following surgical fixation of extracapsular hip fracture with a Gamma nail could be recommended for pain management during walking and functional gait recovery [126].

#### Functional Recovery

Orthogeriatric units can be defined as a transversal and multidisciplinary care model, with the main objective of recovering of previous function in older patients with hip fracture.

Several aspects play a relevant role in the functional recovery after hip surgery in older people. Awareness of the expected recovery following hip fracture is essential for setting of realistic goals. An observational study of 733 patients aged ≥65 years with hip fracture found a low rate of return to previous function, regardless of prefracture functional capacity. Return to independence in activities of daily living (ADLs) was less likely for those >85 years old (20% vs. 44%), with dementia (8% vs. 39%) and with a Charlson comorbidity index greater than 2 (23% vs 44%) [52]

Functional outcomes after a hip fragility fracture seem to depend more on patient characteristics than treatment-related factors [127] In a retrospective cohort study of 519 patients with hip fracture admitted to rehabilitation settings, it has been reported that both delirium and clinical adverse events (infections, respiratory failure, pulmonary embolism, falls) affected functional outcome. A clinical orthogeriatric approach is necessary in order to minimize the impact of these adverse events on the rehabilitation program [128].

A correlation between grip strength measured early after hip fracture and subsequent short and long-term functional recovery was found in a prospective cohort study that included 190 patients. Hand grip weakness was an independent predictor of worse functional outcome 3 and 6 months after hip fracture [129].

Early mobilization after surgery for hip fracture reduces medical complications and mortality. A higher time upright at discharge, measured in the first week after surgery, was associated with less fear of falling, a higher gait speed and a faster Timed Up and Go test time [130].

A single-blind controlled trial reported that a motivational interview conducted with hip fracture patients after being discharged from rehabilitation was related to an increase in physical activity and ambulation capacity [131].

The relationship between specific aspects of the rehabilitation program and functional outcome has been examined in several studies. A randomized controlled trial showed that a hospital rehabilitation program based on the training of specific balance tasks was useful to improve physical function, pain, ADL and quality of life in older patients with hip fracture [132]. Muscle quality (muscle mass and muscle strength) after a hip fracture improved with high-intensity resistance training with the knee in extension in a case series, possibly leading to significant gains in physical function [133].

A systematic review concluded that progressive resistance exercise after hip fracture surgery improved mobility, ADLs, balance, lower extremity strength, and performance task outcomes [134].

## 7. Prognostic Factors and Mortality

Of 2443 patients included in a prospective cohort study included, 36.8% were receiving treatment with β-blocker therapy before surgery. The group treated with beta-blockers was significantly older, had more comorbidities, and was less fit for surgery based on their ASA score; despite these risk factors, 90-day mortality was significantly lower in patients receiving beta-blockers (adjusted incidence rate ratio = 0.82, 95%CI: 0.68 to 0.98, *p* = 0.03) [135].

Preoperative CGA with shared decision-making was compared in a before-after, single-centre, retrospective study. Significantly more patients (or representatives) in the CGA group chose non-surgical management after hip fracture (9.1% vs. 2.7%, *p* = 0.008). Patient characteristics were comparable. Reasons not to undergo surgery included aversion to be more dependent on others and severe dementia [136].

Several studies have researched mortality after hip fracture and its risk factors. Baseline characteristics explained less than two-thirds of the six-month mortality after hip fracture in a retrospective observational study including 1010 individuals (mean age 86 (SD 6) years). The six-month mortality rate was 14.8%. The six-month attributable mortality estimates were as follows: baseline characteristics (including age, gender, comorbidities, autonomy, type of fracture) accounted for 62.4%; perioperative factors (including blood transfusion and delayed surgery) for 12.3%; and severe postoperative complications for 11.9% of attributable mortality [137].

One-year mortality in hip fracture patients from the Nan Province (Thailand) was 19%, or 6.21 times higher than expected compared with the age-matched population. Mortality among hip fracture patients was also significantly higher among those aged older than 80 years, non-ambulatory before the fracture and at hospital discharge, or suffering end-stage renal disease, delirium, and pneumonia [138].

In a retrospective study of 254 patients (mean age, 78.74 years), one-year mortality was 22.8% (58 patients). Univariate analysis identified age >85 years, male gender, ASA score ≥ 3, having ≥3 comorbidities, and a C Reactive Protein to albumin ratio (CAR) ≥ 2.49 were identified as mortality risk factors. The ASA score, CAR and number of comorbidities were included in the binary logistic regression analysis to determine the major predictors of 1-year mortality. The presence of a CAR ≥ 2.49 was found to be a strong indicator for 1-year mortality in patients operated due to hip fracture in the elderly population, while an ASA score ≥ 3 and the presence of ≥3 comorbidities were also related to mortality [139].

A retrospective French cohort study of 309 patients studied risk factors for 1-tear mortality, which was 23.9%. Over half had a surgical delay greater than 48 h (181 patients, 58.6%). Factors independently associated with 1-year mortality were: advanced age (HR = 1.06, 95%CI: 1.01–1.12; *p* =0.032), comorbidities as defined by the revised cardiac index or Lee score ≥ 3 (HR = 1,52, 95%CI: 1,05–2,20; *p* = 0.026) and surgical delay over 48 h (HR = 1.06, 95%CI = 1.01–1.11; *p* = 0.024) [140].

Mortality at one year was 35% and was associated with low IADL day −15 (*p* < 0.01), elevated CIRS-G (*p* < 0.01), severity (*p* = 0.05) and malnutrition (*p* = 0.05) in a prospective study of 113 patients (mean age 87 years (range 76–100). Of those who survived, 45% had a functional decline one year after the fracture and 11% were admitted in a nursing home [141].

The HULP-HF score was designed to predict one-year mortality after hip fractures, using a prospective study of 509 patients with a 1-year mortality of 23.2%. The eight independent mortality risk factors included in the score were age >85 years, baseline functional and cognitive impairment, low body mass index, heart disease, low hand-grip strength, anaemia on admission, and secondary hyperparathyroidism associated with vitamin D deficiency. The AUC was 0.79 for the HULP-HF score, greater than other tools such as the Nottingham Hip Fracture Score (NHFS), ASA classification or Charlson Comorbidity index [142].

Another study evaluated the usefulness of the Hip-MFS (Multidimensional Frailty Score) to predict 6-month all-cause mortality. Secondary outcomes were 1-year all-cause mortality, postoperative complications prolonged hospital stay, and institutionalization. 6-month mortality was 7.3% (35 patients), after a median of 2.9 months (interquartile range 1.4–3.9 months). The fully adjusted hazard ratio per 1-point increase in Hip-MFS was 1.46 (95%CI: 1.21–1.76) for 6-month mortality. The odds ratios for postoperative complications and prolonged total hospital stay were 1.24 (95%CI: 1.12–1.38) and 1.16 (95%CI: 1.03–1.30), respectively. After adjustment, high-risk patients (Hip-MFS > 8) had a higher risk of 6-month mortality (HR: 3.55, 95%CI: 1.47–8.57) than low-risk patients. The Hip-MFS successfully predicted 6-month mortality better than age or other existing tools (*p*-values of comparisons of ROC curves: 0.002, 0.004, and 0.044 for the ASA classification, age and NHFS, respectively). It also predicted postoperative complications and prolonged hospital stay in older hip fracture patients after surgery [143].

## 8. Costs

The acute and post-acute care of patients with an osteoporotic hip fracture pose a significant burden for health care resources all over the world, involving up to 1.5% of total health care budgets [144]. The cost of acute inpatient care of this type of fracture is estimated globally at $13,331 according to a recent systematic review [145]. Costs were significantly associated with prefracture comorbidities prior to fracture and developing a medical or surgical complication during hospitalisation, due to an increase in the length of stay.

The mean cost of hospitalisation of an osteoporotic hip fracture patient was Singapore dollars (SGD) 13,313.81 (€8280,00 at current rates) in a retrospective analysis of patients admitted under a mature orthogeriatric co-management care service in Singapore. The presence of complications significantly increased average cost (SGD 2,689.99 [€1672,93] more than if there were no complications). Each additional day between admission and time of surgery led to an increased cost of SGD 575.89 (€358,15), with surgery after more than 48 h costing an average of SGD 2,716.63 (€1689,50) more than surgery within 48 h. The authors concluded that a standardised co-management model of care could accelerate surgical treatment and help reduce peri- and postoperative complications, reducing overall costs of these fractures [146].

A prospective, 12-month observational study from Spain calculated the mean total cost in the first year after an osteoporotic hip fracture at €9690 (95%CI: 9184–10,197) in women and €9019 (95%CI: 8079–9958) in men. Initial hospitalization was the main determinant of cost, followed by ambulatory care and home care. The cost per day of hospital stay has been estimated at €1,000, so a delay of 1 day for hip surgery would cost approximately 1800 € [144,147].

In addition to the direct costs derived from inpatient acute care, most of the costs for osteoporotic hip fractures are associated with post-acute care, including the direct costs for rehabilitation, medium and long-term care, and the indirect costs related with absence from work of family caregivers [148,149]. All these contribute to total costs reaching $43,669 per patient in the first year after a hip fracture, higher than those estimated for acute coronary syndrome ($32,345) and ischaemic stroke ($34,772) [145].

Many initiatives have been created in order to improve outcomes and reduce costs in an attempt to alleviate this overall burden of health care systems. The implementation of the orthogeriatric co-management model of care, integrated in specific functional units, has been a vital tool to improve outcomes [150].

The implementation of orthogeriatric programs has been shown by several studies to offer greater cost-effectiveness than usual care, decreasing surgical delay, length of stay and improving physical function, with a decrease in one-year morbidity and mortality, while using fewer resources per patient and saving money [151,152], as has also been shown in systematic reviews and meta-analysis that associated orthogeriatric programs with decreases in time to surgery, LOS, complication rates and costs [32,153].

Another recent study evaluated the cost-effectiveness of orthogeriatric models and nurse-led fracture liaison services (FLS), compared with usual care. Orthogeriatric models of care were the most effective and cost-effective models, at a threshold of £30,000 per quality-adjusted life years gained (QALY). The authors concluded that introducing an orthogeriatric model of care and a FLS was cost-effective when compared with usual care, regardless of how patients were stratified in terms of age, sex, and Charlson comorbidity score at the moment of index hip fracture [154].

A systematic review of eight studies (two high-quality and six moderate or low-quality studies) showed that the implementation of Comprehensive Geriatric Assessment (CGA) improved return of function and mortality, with reduced cost. The authors concluded that CGA was the most cost-effective care model for orthogeriatric patients [155].

The effect of orthogeriatric clinical care pathways (OG-CCPs) on physical function and health-related quality of life (HRQoL) following hip fracture was evaluated in a systematic review and meta-analysis that included 22 studies (21 included hip fracture patients, and one included wrist fracture patients; the majority were assessed as high quality). Compared with usual care, the OG-CCP group showed moderate improvements in physical function and HRQoL. Inpatient OG-CCPs that extended to the outpatient setting showed greater improvements compared to those that only included inpatient or outpatient management. OG-CCPs that incorporated a care coordinator, geriatric assessment, nutritional advice, prevention of inpatient complications, rehabilitation, and discharge planning also demonstrated greater improvements in outcomes [32].

Though certain questions remain open regarding which model of care should be considered ideal, implementation of an orthogeriatric co-management model of care, integrated in specific functional units, benefits older patients with hip fractures, improving standards of care in a cost-effective manner. Because of that we undoubtedly recommend developing orthogeriatric units as a standard of care of older patients with this type of fracture [156].

## 9. Future Perspectives and Lines of Research

Some recent publications should be mentioned that evaluate the role of advanced practice nurses in the management of hip fracture patients in reducing length of stay and mortality, as in a systematic review by Allsop et al. that included 19 papers [157]. Nurses could play an important role in the multidisciplinary team, for example as coordinator or case manager, improving bone health assessment and falls prevention programs in Fracture Liaison Services (FLS).

The effect of different models of orthogeriatric care for older hip fracture patients was compared to usual orthopaedic care in a meta-analysis and showed that orthogeriatric care was associated with higher odds of diagnosing osteoporosis, initiation of calcium and vitamin D supplements and discharge on anti-osteoporosis medication, but evidence on fall prevention and subsequent fractures was scarce and inconclusive [158]. Future studies could assess combination of orthogeriatric care and FLS with orthogeriatric care alone. Another area of interest is reducing inequity in research regarding rehabilitation interventions in hip fracture patients. In over half of the trials included in a systematic review, potential participants were excluded based on residency in a nursing home, cognitive impairment, mobility/functional impairment, minimum age and/or non-surgical candidacy [159]. These sources of bias should be avoided in future studies.

An emergent topic for study is the race and sex-related differences in hip fracture outcomes. A review published ten years ago, including an important number of papers from USA [160] showed that men were younger and sicker than women with hip fractures, but mortality in men was twice that of women. African-Americans, as well as Hispanics and Native Americans had higher mortality than Whites. Another recent study from the USA reported a significant disparity in surgical delay and perioperative complication rates between races, with African-Americans having a longer time to surgery than Whites [161]. While sex-related differences in some outcomes have been studied more extensively, race-related differences in outcomes are an interesting line of research especially in countries and regions where socioeconomic factors or other factors related to ethnicity could change the access of individuals to healthcare services.

## 10. Conclusions

The efficiency and benefits of orthogeriatric care in a co-management pathway should be generalized globally. Over the past 70 years, orthogeriatric units have enabled major improvements in the standards of care provided to geriatric patients admitted at hospital due to hip fracture. Increased survival and functional recovery rates have been reported across these years, as well as decreased complications and adverse events during hospitalisation, such as the incidence of infections and geriatric syndromes. All these points have led to a decrease in the length of stay and health and social costs. A large number of clinical trials and meta-analyses published over the last 5 years support this evidence.

Nevertheless, there are still knowledge gaps regarding specific clinical issues. Furthermore, lack of continuity of care after hospital discharge is still common nowadays. Gender- and sex-related differences should be further studies, particularly in regions where they entail differences in access to care. While future studies are needed to help answer these open questions but we could ask ourselves if we should apply the strong evidence available in our routine in the meantime, as well.

## Figures and Tables

**Figure 1 ijerph-18-03049-f001:**
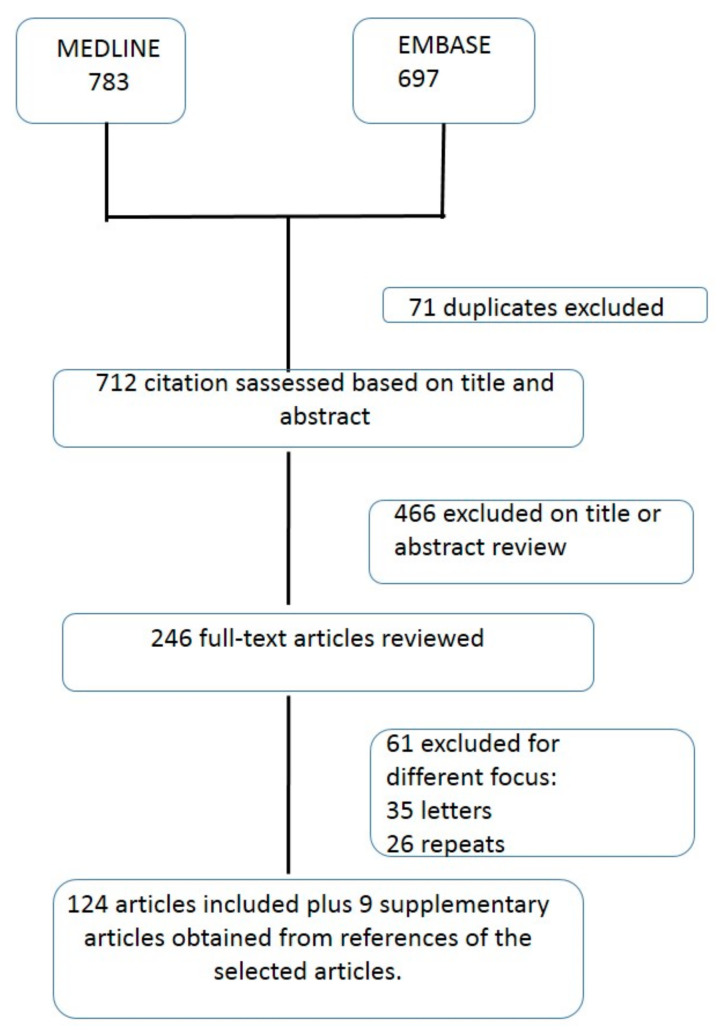
Flow chart of the study selection procedure used in the literature research.

**Table 1 ijerph-18-03049-t001:** Summary of the some most relevant studies on geriatric syndromes and functional recovery included in this review.

Authors	Year	Country	Sample Size	Inclusion Criteria	Exclusion Criteria	Study Design	Conclusion Summary	Level of Evidence
Zusman [61]	2017	Canada	53	Hip fracture aged 65 years or older with a recent hip fracture (3–12 months).	older adults who, prior to the fracture, were unable to walk 10 m, dementia, and/or older adults moved to a residential care facility.	RCT	44% of study participants self-reported UI.	Ib
Morri [65]	2018	Italy	840	65 years of age or older hospitalized	The absence of a legal guardian to sign the consent form in cases of cognitive deficit, and a diagnosis of periprosthetic or pathological fracture	Prospective cohort study	50% sample studied unable to recover their prefracture autonomy levels. Risk factors: increased number of days with diapers (B = 0.003; *p* < 0.001), urinary catheter (B = 0.03; *p* < 0.001)	IIb
Díaz de Bustamante [71]	2017	Spain	509	Patients aged ≥ 65 yo admitted due to hip fracture		Cohort study	81.2% protein malnutrition. 17.1% energy and protein malnutrition. 93% Low vitamin D levels. Sarcopenia prevalence 17.1%	IIb
Inoue [73]	2019	Japan	205	Patients aged ≥ 65 yo, fractures caused by falling and surgical treatment.	Terminal malignant disease, uncontrolled chronic liver disease and/or pre-fracture ambulation difficulty	Longitudinal cohort study	MNA-SF had a significant association with discharge motor-FIM, efficiency on the motor-FIM and 10-m walking speed. GNRI significantly associated with 10-m walking speed.	IIb
Inoue [77]	2017	Japan	204	Age ≥ 65 yo, fractures incurred as a result of falls and required surgery.	Terminal malignant disease, uncontrolled chronic liver disease, and/or pre-fracture ambulation difficulty, partial or no weight-bearing indications during postoperative rehabilitation	Multicentre cohort study	MNA-SF was a significant independent predictor for FIM at discharge (well-nourished vs. malnourished, β = 0.86, *p* < 0.01).	IIb
Beauchamp-Chalifour [78]	2020	Canada	209	Geriatric patients (>65 yo) admitted for a hip fracture.	Subtrochanteric fracture, pathologic hip fracture and polytrauma patients	Cohort study	Deceased patients had lower MNA scores (mean 19.9 (SD 5.2), *p* < 0.001) and lower MMSE scores (mean 16.0 (SD 10.9, *p* < 0.001).	IIb
Torbergsen [79]	2019	Norway	71 patients (31 in the intervention group and 40 controls)	Fracture resulted of a low energy trauma.	Moribund at admittance.	RCT	Intervention group: Vitamin K1 K1: 1.0 (SD 1.2) vs 0.6 (SD 0.6) ng/mL, *p* = 0.09; 25(OH)D: 60 (SD 29) vs 43 (SD 22) nmol/L, *p* = 0.01	Ib
Landi [83]	2017	Italy	127	Age ≥ 70 yo admitted to in-hospital Geriatric Rehabilitation Unit with hip fracture.		Longitudinal cohort study	Sarcopenia 33.9%. Sarcopenia increased risk of incomplete functional recovery: OR 3.07, 95%CI 1.07–8.75. Sarcopenia showed lower Barthel index scores at discharge: 69.2 versus 58.9; *p* < 0.001); and after 3 months of follow-up (90.9 versus 80.5; *p* = 0.02).	IIb
van de Ree [84]	2019	Netherlands	696	Patients ≥ 65 yo with hip fracture	Pathological hip fractures.	Multicentre longitudinal cohort study	53.3% were frail. Frailty was negatively associated with HS (β −0.333; 95%CI −0.366 to −0.299), self-rated health (β −21.9; 95%CI −24.2 to −19.6) and capability well-being (β −0.296; 95%CI −0.322 to −0.270) in elderly patients 1 year after hip fracture. After adjusting for confounders, associations were weakened but remained significant.	IIb
Wei [89]	2017	China	8 studies (22,180 patients)	Types of studies: observational studies; Types of participants: patients with hip fracture; Comorbidity: compared patients with diabetes with those without diabetes		Meta-analysis	Mean PU incidence: 15.1% in group with diabetes compared to 7.5% without diabetes group. Diabetes PU OR 1.825 (95%CI: 1.373–2.425; *p* < 0.001). Subgroup analysis by PU stage: OR 1.474 [95%CI 0.984–2.207] for ≥category II PU, and 2.814 [95%CI: 2.115–3.742] for ≥category I PU.	Ia
Klestil [90]	2018	Austria	28 prospective studies (31,242 patients).	Randomised controlled trials, non-randomised controlled trials, and prospective controlled cohort studies. Adults aged 60 years or older undergoing surgery for acute intra- and extracapsular hip fracture.		Meta-analysis	48 h surgery: RR dying within 12 months (RR) 0.80, 95%CI 0.66–0.97. Adjusted data: fewer complications (8% vs. 17%) in patients who had early surgery.	Ia
Ganizeo [91]	2019	Italy	761	Fragility hip fracture patients aged ≥65 years.	Patients with periprosthetic or pathological fractures.	Prospective cohort study	The incidence of category II or higher PUs was 12%. Five factors independently associated with category ≥II PU development: Higher preoperative Braden score (Hazard Ratio [HR]: 0.884; 95% confidence interval [CI]: 0.806–0.969), surgical procedure with internal fixation (HR 1.876; 95%CI: 1.183–2.975), a higher percentage of days with the presence of foam valve before surgery (HR: 1.010; 95%CI: 1.010–1.023) and a urinary catheter (HR: 1.013; 95%CI: 1.006–1.019) and diaper (HR: 1.007; 95%CI 1.001–1.013) in the postoperative period.	IIb
Chiari [94]	2017	Italy	1083	Patients ≥ 65 years of age with fragility hip fracture.	Patients with periprosthetic or pathological fractures, and patients who presented with pressure ulcers.	Prospective cohort study	Pressure ulcers incidence: 22.7%. Two risk factors: age > 80 years (odds ratio (OR) 1.03; 95%IC 1.006; 1.054, *p* = 0.015), the length of time a urinary catheter was used (OR 1.013; 95%IC 1.008; 1.018, *p* < 0.001.	IIb
Forni [97]	2018	Italy	467			Prospective cohort study	Of these, 27% (*n* = 127) developed a pressure injury. Multivariate analysis identified the following predictive factors: age older than 81 years, type of surgery, and placing the limb in a foam rubber splint.	IIb

## Data Availability

This a review manuscript, we did not need a data availability statement.

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
