# Peer review of "Orthogeriatric Management: Improvements in Outcomes during Hospital Admission Due to Hip Fracture"

_ijerph, 2021, doi:10.3390/ijerph18063049_

Round 1

Reviewer 1 Report

  1. The authors write, “Approximately1.3 million hip fractures were diagnosed in 1990 [1], and this figure is expected to increase to over 6 million by 2050.” Please indicate whether this refers to a global or national figure. Also, is this cumulative prevalence or do the authors mean that the incidence in 2050 is projected to be 6 million?
  2. At the end of the first paragraph, the authors introduce the term “orthogeriatric team.” This should be followed by an operative definition, such as, “which is conceptualized as a multi-disciplinary care team specializing in hip fracture treatment.”
  3. The Results Section begins with the statement, “The efficiency and benefits of orthogeriatric care in a co-management pathway should be generalized globally.” This is a conclusion and should be moved to the appropriate section.
  4. In the section on delirium, the authors note a very large spread in incidence: 4.5 and 41.2%. The authors should attempt to explain this difference. Did settings differ? Were study populations not comparable in terms of age and/or comorbidities? Were definitions of delirium dissimilar across studies?
  5. In the section on cognitive decline, the authors site a number of studies that suggest an indication between this and poor recovery, suggesting that the association is causal. But towards the end of the section, they note that after controlling for a number of confounders, a difference in functional recovery was not detected between people with vs. without cognitive decline. It seems that the cognitive decline in fact signals greater frailty, and when this is accounted for, the association with functional recovery is nullified. Also, the authors state that the associations between cognitive decline and functional recovery justify programs for cognitive stimulation of patients following hip fracture. While I believe cognitive stimulation is always good, I don’t believe the authors have successfully made the argument in this section.
  6. The authors should refer to quality of care standards implemented internationally that specify time-to-surgery following hip fracture.
  7. The manuscript is way too long. The authors should focus on those issues and endpoints unique to orthogeriatric care rather than present an exhaustive list of comorbidities and their treatment, many of which are not unique to this care model. Further, included studies should focus on comparisons of outcomes with vs. without orthogeriatric care.

Author Response

The authors wish to thank the reviewers for their time and effort in reviewing our manuscript, as well as for their insightful comments. We have addressed the issues raised to the best of our ability, and hope the revised manuscript meets the reviewers’ standards.

The reviewers’ comments are replied point-by-point below:

Reviewers’ comments:

Reviewer #1:

  • The authors write, “Approximately1.3 million hip fractures werediagnosed in 1990 [1], and this figure is expected to increase to over6 million by 2050.” Please indicate whether this refers to a global ornational figure. Also, is this cumulative prevalence or do the authorsmean that the incidence in 2050 is projected to be 6 million?

  • Authors’ reply: We have rephrased this sentence to clarify the doubts raised: “Approximately 1.3 million hip fractures were diagnosed in 1990 worldwide [1], and this worldwide annual incidence is expected to increase to over 6 million globally by 2050”.

  • At the end of the first paragraph, the authors introduce the term“orthogeriatric team.” This should be followed by an operativedefinition, such as, “which is conceptualized as a multi-disciplinarycare team specializing in hip fracture treatment.”
    • Authors’ reply: We have added the sentence: “Orthogeriatric care can be defined as the collaboration between orthopedic surgeons and geriatricians to improve hip fracture patient outcomes during hospital admission [8]”. We hope this definition is sufficient.

  • The Results Section begins with the statement, “The efficiency andbenefits of orthogeriatric care in a co-management pathway should begeneralized globally.” This is a conclusion and should be moved to theappropriate section.
    • Authors’ reply: We have moved this phrase to the conclusion section.

  • In the section on delirium, the authors note a very large spread inincidence: 4.5 and 41.2%. The authors should attempt to explain this Did settings differ? Were study populations not comparablein terms of age and/or comorbidities? Were definitions of deliriumdissimilar across studies?
    • Authors’ reply: We have added the statement “this wide range in the incidence reported is due to the different ages of the patients included in the studies, the screening tools employed, the types of settings and the surgical and anesthetic techniques used.” to the manuscript.

  • In the section on cognitive decline, the authors site a number ofstudies that suggest an indication between this and poor recovery,suggesting that the association is causal. But towards the end of thesection, they note that after controlling for a number of confounders,a difference in functional recovery was not detected between peoplewith vs. without cognitive decline. It seems that the cognitivedecline in fact signals greater frailty, and when this is accountedfor, the association with functional recovery is nullified. Also, theauthors state that the associations between cognitive decline andfunctional recovery justify programs for cognitive stimulation ofpatients following hip fracture. While I believe cognitive stimulationis always good, I don’t believe the authors have successfully made theargument in this section.
    • Authors’ reply: We have formulated this paragraph more clearly, and hope the argument is better understood in this version. We have furthermore added a recent citation with the phrase “Likewise, the goals of rehabilitation in hip fracture patients with dementia should focus not only on functional recovery, but rather add other objectives such as quality of life, decrease in the complication rate or optimization of social support.”

  • The authors should refer to quality of care standards implementedinternationally that specify time-to-surgery following hip fracture.
    • Authors’ reply: We have cited the NICE quality of care standards specifically regarding surgical delay, which are considered standard internationally. We did not perform a through review of the different quality of care standards regarding surgical delay for brevity.

  • The manuscript is way too long.
    • Authors’ reply: We have substantially shortened the manuscript.

  • The authors should focus on those issues and endpoints unique toorthogeriatric care rather than presentan exhaustive list of comorbidities and their treatment, many of whichare not unique to this care model. Further, included studies shouldfocus on comparisons of outcomes with vs. without orthogeriatric care.
    • Authors’ reply: We have centred the text on the issued suggested, eliminating the other elements not directly related to orthogeriatric care.

Reviewer 2 Report

Thank you for giving me the opportunity to review the manuscript: “Orthogeriatric Care: Reviewing Improvements in Outcomes”. Although this review deals with an interesting topic, the following issues should be addressed, before considering it for publication on IJERPH.
TITLE: please consider a title revision
METHODS: 
- Please add a paragraph dealing with the data extraction process.
- Please add a paragraph focusing on the quality assessment of the studies included in tables 1, 2 and 3. The information about study quality should be also added in tables 1, 2 and 3. 
RESULTS:
Please add a paragraph dealing with gender-related and race-related differences in the outcome of hip fractures. These two topics should be also discussed in the following sections: “Future perspectives and lines of research” and “Conclusion”.
TABLES: please reduce the text in each row, to improve the readability of the tables 
REFERENCES:  limit the number of self-citations to two!
GENERAL COMMENT: consider a moderate English revision.

Author Response

Reviewer #2

Thank you for giving me the opportunity to review the manuscript:
“Orthogeriatric Care: Reviewing Improvements in Outcomes”. Althoughthis review deals with an interesting topic, the following issuesshould be addressed, before considering it for publication on IJERPH.

  • TITLE: please consider a title revision
    • Authors’ reply: We have changed the title to: “Orthogeriatric management: Improvements in Outcomes during hospital admission due to hip fracture” and hope the reviewer agrees with this suggestion.

  • METHODS:

Please add a paragraph dealing with the data extraction process.

  • Authors’ reply: We have added a paragraph explaining the data extraction process.

Please add a paragraph focusing on the quality assessment of the studies included in tables 1, 2 and 3. The information about study quality should be also added in tables 1, 2 and 3.

  • Authors’ reply: We have added a sentence explaining the quality assessment of the studies included.

  • RESULTS:

Please add a paragraph dealing with gender-related and race-related differences in the outcome of hip fractures.

  • Authors’ reply: We agree this is indeed a very interesting topic and have added a paragraph: “An emergent topic for study is the race and sex-related differences in hip fracture outcomes. A review published ten years ago, including an important number of papers from USA [158] showed that men were younger and sicker than women with hip frac-tures, but mortality in men was twice that of women. African-Americans, as well as his-panics and native americans had higher mortality than whites. Another recent study from the USA reported a significant disparity in surgical delay and perioperative com-plication rates between races, with african-americans having a longer time to surgery than whites .[159] While sex-related differences in some outcomes have been studied more extensively, race-related differences in outcomes are an interesting line of research especially in countries and regions where socioeconomic factors or other factors related to ethnicity could change the access of individuals to healthcare services.”

These two topics should be also discussed in the following sections: “Future perspectives and lines of research” and “Conclusion”.

  • Authors’ reply: We have added the paragraph in the “Future perspectives and lines of research” section as well as in the conclusion section.

  • TABLES: please reduce the text in each row, to improve the readability
    of the tables.
    • Authors’ reply: We have edited the tables and reduced the text in each row to improve readability.

  • REFERENCES:  limit the number of self-citations to two!
    • Authors’ reply: We have significantly limited the number of self-citations.

  • GENERAL COMMENT: consider a moderate English revision.
    • Authors’ reply: We thank the reviewer for his thorough review. A native English speaker experienced in scientific writing has reviewed the entire manuscript.

Round 2

Reviewer 1 Report

The authors have successfully addressed the concerns of the reviewer.

Reviewer 2 Report

All the required corrections have been made. The paper is suitable of publication on IJERPH, in the present form.